# Comprehensive and Accurate Molecular Profiling of Breast Cancer through mRNA Expression of *ESR1*, *PGR*, *ERBB2*, *MKI67*, and a Novel Proliferation Signature

**DOI:** 10.3390/diagnostics14030241

**Published:** 2024-01-23

**Authors:** Anne-Sophie Wegscheider, Joanna Gorniak, Sara Rollinson, Leanne Gough, Navdeep Dhaliwal, Agustin Guardiola, Anna Gasior, Denise Helmer, Zoe Pounce, Axel Niendorf

**Affiliations:** 1MVZ Prof. Dr. Med. A. Niendorf Pathologie Hamburg-West GmbH, Institute for Histology, Cytology and Molecular Diagnostics, Lornsenstr. 4, 22767 Hamburg, Germanyd.helmer@pathologie-hh-west.de (D.H.); 2APIS Assay Technologies Ltd., Second Floor, Citylabs 1.0, Nelson Street, Manchester M13 9NQ, UK

**Keywords:** APIS Breast Cancer Subtyping Kit, breast cancer, clinical performance, *ERBB2*, HER2-low, *ESR1*, *PGR*, *MKI67*, FFPE, molecular subtyping, RT-qPCR

## Abstract

Background: An accurate status determination of breast cancer biomarkers (ER, PR, HER2, Ki67) is crucial for guiding patient management. The “gold standard” for assessing these biomarkers in FFPE tissue is IHC, which faces challenges in standardization and exhibits substantial variability. In this study, we compare the concordance of a new commercial RT-qPCR kit with IHC in determining BC biomarker status. Methods: The performance was evaluated using 634 FFPE specimens, which underwent histological analysis in accordance with standard of care methods. HER2 2+ tumors were referred to ISH testing. An immunoreactive score of ≥2/12 was considered positive for ER/PR and 20% staining was used as a cut-off for Ki67 high/low score. RT-qPCR and results calling were performed according to the manufacturer’s instructions. Results: High concordance with IHC was seen for all markers (93.2% for ER, 87.1% for PR, 93.9% for HER2, 77.9% for Ki67 and 80.1% for proliferative signature (assessed against Ki67 IHC)). Conclusions: By assessing the concordance with the results obtained through IHC, we sought to demonstrate the reliability and utility of the kit for precise BC subtyping. Our findings suggest that the kit provides a highly precise and accurate quantitative assessment of BC biomarkers.

## 1. Introduction

In clinical practice, the molecular subtyping of breast cancer (BC) relies on immunohistochemistry (IHC) to measure the protein expression of estrogen receptor ER/*ESR1*, progesterone receptor PR/*PGR*, human epidermal growth factor receptor 2 HER2/*ERBB2*, and marker of proliferation Ki67/*MKI67* [1]. By providing prognostic and predictive information, these biomarkers play a critical role in optimizing treatment outcomes [2,3,4,5].

Concerns regarding the accuracy and reproducibility of IHC testing persist due to high inter-and intra-laboratory variability, with ER and HER2 testing yielding up to 21.4% false-negative and up to 14.5% false-positive results, respectively [6]. To enhance reliability, guidelines have been introduced for HER2 and ER/PR testing [7,8], but challenges remain with the Ki67 marker’s inconsistent assessment, hindering its precise, routine diagnostic use for classifying breast carcinomas into Luminal A and Luminal B subtypes. Variability, even when using the same antibody and when assessed by the same pathologist, underscores the need for standardization in Ki67 evaluation [9], as misclassification can have significant clinical implications. Consequently, alternative assays that offer enhanced robustness, standardization, and precision in assessing the expression of Ki67, HER2, ER, and PR are of benefit.

To address this need, the APIS BC Subtyping Kit has been developed. This in vitro molecular diagnostic test employs RT-qPCR to precisely determine the mRNA expression levels of *ESR1*, *PGR*, *ERBB2* and *MKI67*. By determining the status (positive/high or negative/low) of these crucial markers, the kit accurately classifies the tumor into a specific molecular subtype. The kit also detects three additional proliferative markers, generating a proliferative score encompassing all cell cycle stages. Data analysis is performed using accessible web-based software, presenting a clear and informative output for all marker results. With automated result calling, the need for pathologists to interpret results is reduced, thereby alleviating their increasing workload [10]. Consequently, this streamlined process facilitates faster clinical decision-making for oncologists.

Herein, we report the development and validation of the APIS BC Subtyping Kit, assessing clinical performance, reliability, and reproducibility across numerous testing sites. The APIS BC Subtyping Kit represents a promising advancement towards standardized, precise, and reliable assessment of ER, PR, HER2, and Ki67 expression, holding the potential to improve breast cancer diagnostic pathways.

## 2. Materials and Methods

### 2.1. Clinical Performance Evaluation

#### 2.1.1. Specimen Collection

A total of *n* = 920 invasive BC formalin-fixed paraffin-embedded (FFPE) specimens obtained through core needle biopsy (CNB) or resection were collected in two cohorts (cohort 1 N = 650; cohort 2 N = 270) at a single institution (MVZ Prof. Dr. med. A. Niendorf Pathologie Hamburg-West GmbH) and processed in accordance with approval from the Ethics Committee (EC) of the Hamburg Medical Association (Ethics Approval Vote PV2946). All specimens were collected from different patients.

#### 2.1.2. Pathological Evaluation

Tumor grading, tumor typing, and immunohistochemistry (ER, PR, HER2, Ki67) (IHC) were performed for all specimens by MVZ Prof. Dr. med. A. Niendorf Pathologie Hamburg-West GmbH. IHC was performed and analyzed by two board-certified pathologists according to the institute’s standardized protocols and observing the ASCO/CAP guidelines [7,8]. Tumors with an HER2 score of 2+ were referred to in situ hybridization (ISH) to determine *ERBB2* amplification, with the ISH result replaced the IHC result in determining the HER2 status (positive/negative) of the tumor. Tumors with an immunoreactive score (IRS) of ≥2/12 for ER/PR by Remmele and Stegner were considered positive. The score was determined by factoring in both the percentage of positive cells and the intensity of the staining reaction [11]. Ki67 expression was considered high when ≥20% of the nuclei stained positively.

#### 2.1.3. RNA Extraction

Total RNA was extracted from an at least 6 µm FFPE tissue section with a tumor content ≥ 20%, using the RNeasy^®^ DSP FFPE Kit (QIAGEN, Hilden, Germany) following the manufacturer’s instruction. The RNeasy DSP FFPE Kit was validated by APIS as an RNA extraction method to be used in conjunction with the APIS BC Subtyping Kit. For each specimen, RNA within the eluate was fluorometrically quantified using Qubit 4 Fluorometer (Thermo Fisher Scientific, Waltham, MA, USA), normalized to 2.5 ng/µL and stored at −80 °C until use.

#### 2.1.4. Gene Expression by RT-qPCR

The mRNA expression levels of *ESR1*, *PGR*, *ERBB2*, and *MKI67*, as well as three additional targets encompassing the proliferative signature (*CCNA2*, *KIF23*, and *PCNA*) and two reference genes (*IPO8* and *PUM1*), were determined by RT-qPCR using the APIS BC Subtyping Kit (APIS Assay Technologies, Manchester, UK).

The APIS BC Subtyping Kit comprises four reaction mixes, each consisting of up to three assays (pairs of primers and a probe specific to the respective target sequence). Each assay is labeled with target-specific fluorophores, allowing for the simultaneous detection of all targets. RT-qPCR was performed using the QuantStudio™ 5 Dx real time PCR system (QS5™Dx; Thermo Fisher Scientific). An instrument-specific locked run template was used to ensure that correct reaction settings were applied, as per the APIS BC Subtyping Kit Instructions for Use.

Each patient specimen was analyzed with each assay in duplicate using 10 ng of RNA per reaction. Each RT-qPCR included up to 10 patient specimens and 1 replicate of positive and negative control per assay.

Normalized gene expression levels (ΔCt) for each specimen were determined by subtracting the average cycle threshold (Ct) value of the duplicate measurements of the target of interest from the mean Ct value of the duplicate measurements of the reference genes.

Binary target calls (positive/negative) are based on clinically validated target- and device-specific ΔCt cut-off values. ΔCt cut-off values were established by employing a binary classifier logistic regression model trained using the ΔCt expression data as input and the IHC status input as the binary classifier. The value that achieved an equal balance between the sensitivity and specificity of the assay, and demonstrated the highest performance, was chosen as the ΔCt cut-off for each target.

A logistic model using ΔCt values of *MKI67*, *CCNA2*, *PCNA*, and *KIF23* was used to calculate a proliferation score (between 0 and 1); a value less than 0.5 was reported as low proliferation, and greater than 0.5 as high proliferation. The combination of *ESR1*, *PGR*, *ERBB2*, and *MKI67* status was used to determine the molecular subtype, based on a definition by the St Gallen working group [12] (Table 1).

#### 2.1.5. Statistical Analysis

The run validity and result calling were performed using APIS BC Subtyping Kit analysis software (APIS Assay Technologies). The software uses the QS5 Dx result file, providing run and sample validity information, normalized gene expression levels (ΔCt), binary marker status, proliferation measurements, and tumor subtype.

The agreement between the APIS BC Subtyping Kit and the reference method was evaluated by constructing 2 × 2 cross-tables, and calculating the overall percent agreement (OPA), positive percent agreement (PPA; diagnostic sensitivity), negative percent agreement (NPA; diagnostic specificity), positive predictive value (PPV), and negative predictive value (NPV), alongside their respective two-sided 95% confidence intervals (95% CIs). JMP 16.1.0 software (SAS^®^ Institute) was used for all statistical analyses.

### 2.2. Analytical Precision and Reproducibility

#### 2.2.1. Study Design

The precision of the APIS BC Subtyping Kit was determined in accordance with the standard methods provided in CLSI EP05-A3 guidelines [13]. The study was carried out across three testing sites: APIS Assay Technologies (site 1), and two independent molecular laboratories—MDNA Life Sciences UK Ltd. (Site 2; Gateshead, UK) and The North East Innovation Lab, Newcastle upon Tyne Hospitals NHS Foundation Trust (site 3). Prior to the study start, all operators were trained and deemed proficient in the use of the APIS BC Subtyping Kit. During the study, each site performed repeated measurements with the APIS BC Subtyping Kit according to the predefined study plan. At each site, two different APIS BC Subtyping Kit lots were tested, with a total of three kit lots used across all sites. The study comprised five non-consecutive days for sites 2 and 3, and 10 non-consecutive days for site 1, resulting in a generation of 100 measurements per sample. Full details of the study design can be found in Appendix A.

#### 2.2.2. Samples

The samples used in this study were prepared by extracting total RNA from clinical FFPE breast cancer tissue blocks, as described above. A total of six RNA samples were contrived, representing negative, low-positive and mid-positive expression levels for each target. Samples were divided into single use aliquots and distributed to all study sites.

#### 2.2.3. Gene Expression by RT-qPCR

The mRNA expression levels of each target were measured as described above. Each sample was measured in a single replicate. Binary target calls were determined for each measurement.

#### 2.2.4. Statistical Analysis

The precision and reproducibility of the APIS BC Subtyping Kit were assessed on binary (single marker status) and semi-quantitative (∆Ct) levels. To assess the precision of single-marker status calling (positive/high; negative/low), the proportion of results in agreement with the expected target call was reported alongside corresponding 95% CIs. The output of the proliferative signature was also used to summarize the hit rate for the proliferative score, using Ki67 status as a reference.

For quantitative precision assessment, variance component analysis was carried out using JMP 16.1.0 software (SAS Institute Inc., Cary, NC, USA), utilizing main effects models and the Restricted Maximum Likelihood (REML) estimates only. The factors were nested in the following order: Site, Operator, Instrument, Lot, and Run/Day. The variability associated with each factor was reported in terms of standard deviation (SD), and the percent of total variance, where the variability corresponding to each component is calculated as a percentage out of the sum of all variance components. Estimates of reproducibility were based on the analysis of the full dataset (three sites). Repeatability was generated using replicate measurements of the same samples using an identical PCR layout, with two operators and two instruments, at Site 1 over 10 days.

## 3. Results

### 3.1. Clinical Performance Evaluation

#### 3.1.1. Patient Population

Out of the 920 patient specimens, 229 were excluded from further analysis due to not meeting inclusion criteria (the reasons for withdrawal are detailed in Appendix A). The inclusion criteria consisted of confirmed invasive breast cancer specimens (CNB or resected FFPE tissue) with >20% tumor content, not older than 18 months, and with documented ethics and informed consent or a waiver for research. An additional 54 specimens were excluded due to insufficient RNA. Testing with the APIS BC Subtyping Kit was conducted at two different locations: APIS Assay Technologies (259 specimens; cohort 2) and MVZ Prof. Dr. med. A. Niendorf Pathologie Hamburg-West GmbH (378 specimens; cohort 1) (Figure 1).

The characteristics of cohort 1 and cohort 2 patient populations were broadly similar for histological type and grade in order to keep the confounding factors to a minimum. Cases included invasive breast cancers of the histological subtypes NST (no special type, ductal carcinoma) and lobular carcinoma, of low, intermediate, or high grade, with or without expression of the estrogen and/or progesterone receptor, of HER2 positive or negative status, and of high or low proliferation rate (Ki67) to avoid sample bias. Due to a low prevalence in the study population, HER2-positive tumors make up the smallest group of patients in cohort 1. Cohort 2 comprised only CNB specimens. Within cohort 1, 62.4% of specimens were collected via CNB and 37.6% via resection. The basic clinicopathological characteristics of the tumors used for analysis are listed in Table 2.

#### 3.1.2. Agreement between APIS Breast Cancer Subtyping Kit and IHC

The overall agreement between mRNA expression levels determined by APIS BC Subtyping Kit and IHC is shown in Table 3. A strong correlation between ∆Ct values and the percentage of staining with IHC was observed for *ESR1*/ER (R^2^ = 0.664, *p* < 0.0001) (Figure 2A) and *PGR*/PR (R^2^ = 0.663, *p* < 0.0001) (Figure 2B). A moderate level of correlation was noted between *MKI67* ΔCt and the percentage of staining with IHC (R^2^ = 0.414, *p* < 0.0001) (Figure 2C), and between Ki67 IHC staining and proliferation score (R^2^ = 0.472, *p* < 0.0001) (Figure 2D). A clear stratification of ΔCt by HER2 IHC status was noted (Figure 3). Patients assigned HER2 2+ status were classified into negative and positive groups by the APIS BC Subtyping Kit in line with the results obtained from ISH testing. This result is of utmost importance, as the anti-HER2 neo-adjuvant therapy decision is based on the results provided by this test. The APIS BC Subtyping Kit accurately detects HER2 expression (with IHC 2+/ISH+ and IHC 3+ cases defined as positive). *ERBB2* mRNA expression was detected by the APIS BC Subtyping Kit in a subset of patients with 0 and 1+ IHC HER2 scores, highlighting the continuous nature of *ERBB2* expression, and providing an opportunity to enhance HER2 stratification into a HER2-low category. Notably, the majority of cases showing disagreement between ER and PR calls made by APIS BC Subtyping Kit and IHC status were found to be in close proximity to their respective ΔCt cut-off values. Considering the binary nature of positive and negative results for ER and PR and the inherent variability of quantification methods, the biological implications of this discordant calling require further investigation.

#### 3.1.3. Subtype Calling Agreement

For each tumor, the subtype call by the APIS BC Subtyping Kit was compared to the subtype assigned by the pathologist, based on the IHC/ISH test results. A high overall percent agreement for every subtype was shown (71.59%). It is important to note that this result is most likely driven by the differences in *MKI67* RNA and Ki67 protein expression. Indeed, when taking into consideration ER, PR, and HER2 status only, the overall percent agreement in subtype calls was shown to be notably higher (89.56%). The PPA, NPA, and OPA for each subtype, along with 95% CIs, are shown in Table 4 and the full breakdown of subtype calling by each method is displayed in Figure 4.

#### 3.1.4. Agreement between CNB and Resected FFPE Specimens

The clinical performance of the APIS BC Subtyping Kit was assessed with both sample types: CNB (N = 492) and resection (N = 142). All specimens were collected from different patients. Overall, an agreement of over 85% was achieved for both CNB and resected sample types for ER, PR, and HER2. Performance was comparable for the two sample types, across all markers (Table 5).

### 3.2. Analytical Precision and Reproducibility

#### 3.2.1. Analytical Precision

Analytical precision was calculated, taking into consideration data generated by Site 1. The variability attributed to factors such as operator, instrument, kit lot, and run (including between-day variability) was assessed and reported as standard deviation (SD) and as a percentage of the total variance. These percentages represent the contribution of each factor to the overall variance (Table 6).

#### 3.2.2. Inter-site Reproducibility

Between-site reproducibility was calculated across all three sites involved in the study. The primary source of variability was the replicates within each run, while inter-instrument variance had minimal impact on assay variation. The between-site analysis revealed excellent reproducibility in quantitative measurements, with a total standard deviation (SD) ranging from 0.00 to 0.22 for positive samples (Table 7).

#### 3.2.3. Binary Single Marker Agreement

The proportion of results in agreement with the expected result was reported along with the corresponding 95% CIs displaying excellent concordance (Table 8); 100% reproducibility was achieved for ER, PR, and HER2 for all tested samples (negative/low-positive/mid-positive), and for most Ki67 samples (87% agreement for negative samples).

## 4. Discussion

The accurate and reliable determination of ER, PR, HER2, and Ki67 expression is crucial for precise breast cancer diagnosis and effective patient management. However, the reliability of current “gold standard” methods such as IHC, particularly for Ki67, is limited due to significant variations observed within and between laboratories [9]. To overcome these limitations, the APIS BC Subtyping Kit aims to determine the status of these markers accurately and consistently, enabling the delivery of precise and effective patient management decisions.

Indeed, the APIS BC Subtyping Kit has demonstrated a strong agreement with IHC for all markers, exhibiting excellent accuracy and repeatability. Furthermore, its performance across multiple testing sites highlights the potential to enhance the reproducibility of assessing these markers in a diagnostic setting. The ability of the APIS BC Subtyping Kit to achieve highly repeatable and reproducible results is advantageous compared to current methods that rely on IHC and are prone to inconsistencies. The APIS BC Subtyping Kit minimizes the reliance on highly trained and experienced personnel for accurate results interpretation, thus improving the efficiency and reliability of breast cancer management.

Our study’s limitations include its retrospective design, relying on specimens procured from a singular institution across two cohorts. Despite this, no substantial differences were noted between these cohorts. Cohort one was curated to maintain an even representation of all BC subtypes, whereas cohort two aimed to mirror the real-world distribution of BC subtypes, inadvertently yielding a greater number of HER2-negative specimens compared to HER2-positive specimens.

Here, we found that the APIS BC Subtyping Kit exhibited a diagnostic sensitivity (PPA) of over 90% for *ESR1*, *PGR*, and *ERBB2*. We found that the discrepancies were mainly seen in tumors with RNA levels close to the ∆Ct cut-offs and lower protein expression detected by IHC. This finding underscores the significance of employing semi-quantitative methods in result reporting, especially since previous studies indicated that low-ER-expressing tumors often present characteristics more similar to ER-negative cancers. As is still not clear what clinical implications these discrepancies may have, it is important to incorporate clinicopathological characteristics, such as grade, number of nodes, and tumor size, in patient management decisions [14]. Further explorations of the links between semi-quantitative mRNA expression and treatment response are required to demonstrate its clinical value for therapeutic decision-making.

Higher discordance was observed for *PGR*/PR measurements, which aligns with the heterogeneous distribution of PR staining and the resulting increased variability across different tissue sections [15]. These discrepancies are likely attributed to measurement uncertainty and tumor heterogeneity. It is important to note that due to the nature of the assay, RT-qPCR was performed on a separate section that, although spatially close to the section used for IHC, inherently exhibits heterogeneity. Alternative RT-qPCR-based kits already available on the market utilizing RNA extracted from the same tissue section could be used for comparison, helping to address potential discrepancies arising from tumor heterogeneity.

The agreement between IHC and RT-qPCR for *MKI67*/Ki67 was moderate with a diagnostic sensitivity (PPA) of 79.86%, indicating that expression at the RNA level does not always translate to observable protein by IHC. However, Ki67 staining reproducibility has been widely discussed in the literature [9,16], with significant inter- and intra-laboratory variability observed for this marker. Challenges of IHC processing and analytical issues in scoring IHC slides have resulted in a “Grey Zone” for Ki67 scoring being suggested, where only IHC values below 5% or above 30% should be considered as low or high expression, respectively, as reproducibility within this range is particularly poor [16]. A substantial overlap in RT-qPCR results between tumors categorized as high and low with IHC can be observed, although the populations are clearly distinct, as higher ΔCt values correspond to increased IHC % staining. Consistent with previous studies, most of the discordant cases for this marker were identified as false positive (i.e., false high) cases [15].

It is worth noting that Ki67 IHC analysis might selectively focus on staining hotspots during image analysis, whereas RT-qPCR evaluates the average *MKI67* score across an entire FFPE section [17]. This averaging process across the section may contribute to the differences observed between mRNA and IHC results.

The current routine diagnostic assessment for BC markers involves the use of IHC, and when necessary, additional reflex testing for HER2 using ISH [8]. However, in the clinical setting, there is a clear need for a rapid and accurate testing method. Notably, the APIS BC Subtyping Kit provides a single-resolution *ERBB2*/HER2 detection method, removing the requirement for additional reflex testing and thus providing valuable information to healthcare providers in a rapid manner, reducing overall turnaround time and aiding fast patient management decisions. As demonstrated in Figure 3, the APIS BC Subtyping Kit correctly assigned positive status to 6 out of 7 HER2 2+ ISH positive tumors, and negative status to 57 out of 62 HER2 2+ ISH negative tumors.

The quantification of HER2-low in breast cancer has garnered considerable interest due to emerging evidence of the efficacy of targeted therapies in this patient subgroup [18]. Trastuzumab deruxtecan (T-Dxd), an antibody-drug conjugate (ADC) targeted at HER2, has recently gained approval in the USA and Europe for treating HER2-low breast cancer, which is currently defined as IHC scores of 1+ or 2+ without *ERBB2* ISH amplification. However, data from the DAISY phase II trial reported a response to T-Dxd treatment of 29.7% in tumors with HER2 IHC 0 score (vs. 37.5% in HER2-low patients) [19]. In fact, our dataset shows a substantial overlap in *ERBB2* expression across all HER2 “negative” tumors, including those classified as HER2 IHC 0 score, highlighting the continuous nature of *ERBB2* expression and higher sensitivity of RT-qPCR-based detection approaches. Furthermore, these results confirm that IHC stratification may not be an appropriate method for predicting the response to novel anti-HER2 therapies, such as T-Dxd. Understanding the correlation between APIS BC Subtyping Kit *ERBB2* mRNA expression and the response to anti-HER2 treatments could aid in stratifying patients into responder and non-responder groups, potentially reducing unnecessary severe side effects and offering a more personalized treatment approach. The implementation of additional ∆Ct cut-offs could allow the further stratification of *ERBB2* expression into negative, low, and high status; however, in order to validate this approach, additional studies showing a correlation of the expression level to anti-HER2 treatment response would be required.

The inclusion of the novel proliferation signature within the kit presents a potential avenue for advancing the clinical utility of *MKI67* in the measurement of tumor proliferation. Proliferation status is a commonly used factor in determining the need for chemotherapy in patients with luminal breast cancer, in cases where patients have low-risk features such as node status [18]. The inclusion of three additional targets (*CCNA2*, *PCNA*, and *KIF23*) within the APIS BC Subtyping Kit, known to be associated with breast cancer proliferation [20,21,22,23,24] and expressed across all cell cycle stages, allows for the generation of a more complete proliferation measure. While this score is correlated with Ki67 staining, establishing a correlation between RNA measurement and clinical outcome would provide valuable information. The ability to consistently determine tumor proliferation would be a valuable addition to the pathology evaluation of breast cancer patients, providing clinicians with a reliable measure of proliferation. Various multigene tests addressing this issue exist. Oncotype Dx^®^ (Exact Sciences, Madison, WI, USA), Prosigna^®^ (NanoString Technologies, Seattle, WA, USA), EndoPredict^®^ (Myriad Genetics, Salt Lake City, UT, USA), and MammaPrint^®^ (Agendia, Irvine, CA, USA) provide risk scores with prognostic information for distant recurrence in patients with Luminal A and Luminal B tumors. The risk determined by these tests (high/low) is often used to guide chemotherapy treatment decisions [25]. However, these tests are often centralized and very expensive, and therefore only used as an additional test for those cases that require further assessment.

The exploration of the prognostic potential of the APIS BC Subtyping Kit’s proliferation signature by testing its utility in predicting treatment outcomes and identifying subtypes in comparison to the commercially available tests will be important to provide a centralized and accessible cost-effective alternative.

Subtype calls by the APIS BC Subtyping Kits demonstrated an overall percent agreement for every subtype of over 75%. The lowest performance was identified in distinguishing Luminal B HER2- and Luminal A subtypes. The primary distinction between these two subtypes revolves around the expression level of Ki67. As previously discussed, Ki67 IHC results should be interpreted with caution; rather than distinct high and low populations, Ki67 expression (both at the protein and mRNA level) shows a considerable overlap (Figure 2C). Given this uncertainty in distinguishing between Ki67 high and low expression, some ambiguity should also be anticipated when differentiating Luminal B HER2- and Luminal A tumors. Different breast cancer subtypes exhibit different clinicopathological features, recurrence patterns, and survival outcomes. Molecular subtyping, based on biomarker status, can provide clinical information that could be used to facilitate breast cancer management decisions.

Once validated, the APIS BC Subtyping Kit could provide information on individual biomarker status, BC subtype, and recurrence risk in one test. This would distinguish it from the aforementioned tests, which do not always report individual biomarker status, and therefore cannot be used for the initial biomarker assessment. Future studies are required to compare the subtype calls generated by the APIS BC Subtyping test and other molecular classifiers (e.g., the Prosigna test).

The APIS Breast Cancer Subtyping Kit can provide information about the predictive and prognostic power of the examined parameters, especially on small biopsy material in, e.g., neoadjuvant settings with planned cytotoxic chemotherapy or concerning proliferation rate in endocrine challenge (neoadjuvant endocrine therapy).

## 5. Conclusions

The APIS Breast Cancer Subtyping Kit provides a safe, highly accurate, and reproducible method for the assessment of breast cancer biomarkers and molecular subtypes in patients with invasive breast cancer, using resected and CNB FFPE tissues. The kit proved its reliability when used by clinical scientists in real clinical settings, exhibiting potential to improve the quality of primary breast cancer diagnostics. By providing results in a timely manner, the APIS BC Subtyping Kit holds potential to improve routine practice, by saving pathologists time. Results of immunohistochemistry could be validated and controlled (also for the purpose of quality assurance) by the APIS BC Subtyping Kit or even be replaced in settings where routine IHC diagnostic is not feasible or where results could be provided faster using the APIS BC Subtyping Kit.

Further clinical investigations are required to validate the APIS BC Subtyping Kit’s prognostic potential and establish appropriate thresholds for the semi-quantitative evaluation of *ERBB2*/HER2. The reproducibility and semi-quantitative capabilities of the APIS BC Subtyping Kit could offer researchers a valuable opportunity to uncover meaningful associations between marker expression and treatment response, potentially improving personalized management strategies for breast cancer patients.

## Figures and Tables

**Figure 1 diagnostics-14-00241-f001:**
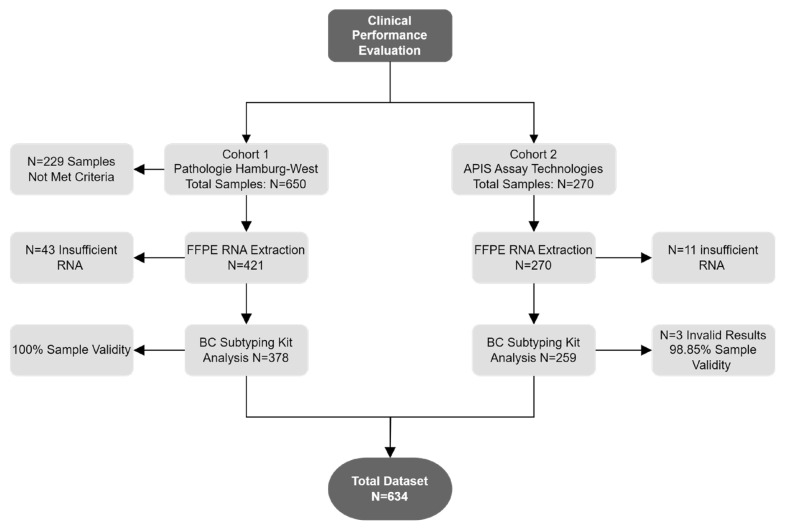
Schematic of specimens used for clinical performance evaluation.

**Figure 2 diagnostics-14-00241-f002:**
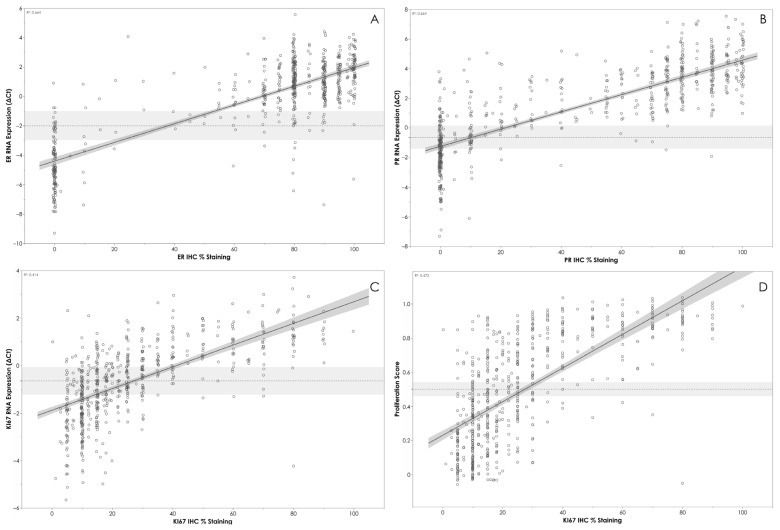
(**A**) APIS BC Subtyping Kit results (ΔCt) for ER plotted against ER IHC %. (**B**) APIS BC Subtyping Kit results (ΔCt) for PR plotted against PR IHC %. (**C**) APIS BC Subtyping Kit results (ΔCt) for KI67 plotted against KI67 IHC %. (**D**) APIS BC Subtyping Kit Proliferation Score plotted against KI67 IHC %. Individual target cut-offs are indicated by the dashed line. Shading around cut-off lines indicates assay’s precision. The majority of discordant calls are close to the assay cut-off.

**Figure 3 diagnostics-14-00241-f003:**
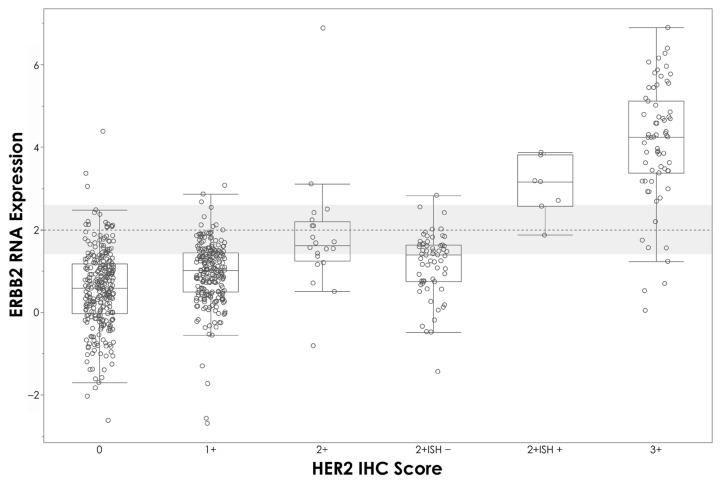
APIS BC Subtyping Kit results (ΔCt) plotted against HER2 IHC score. HER2 cut-off is indicated by the dashed line.

**Figure 4 diagnostics-14-00241-f004:**
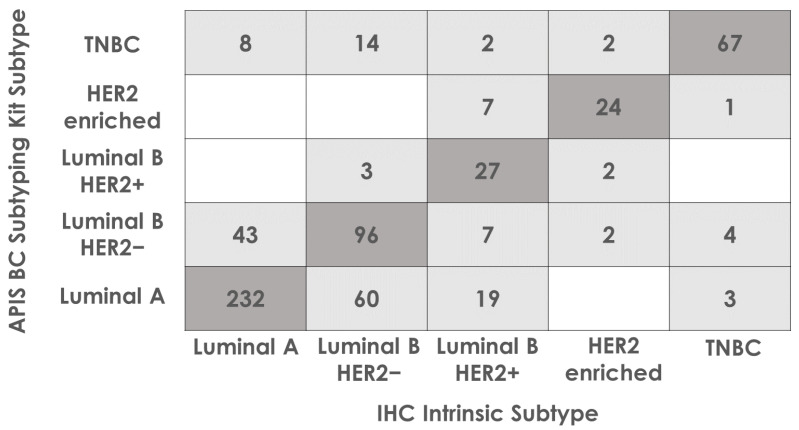
Subtype calling agreement between APIS BC Subtyping Kit and the intrinsic subtype determined by the pathologist based on IHC results. Eleven cases have been removed due to having no assigned IHC subtype. Cases where APIS BC Subtyping Kit calls agreed with IHC intrinsic subtypes are highlighted in dark grey. The remaining cases are highlighted light grey.

**Table 1 diagnostics-14-00241-t001:** Subtype calling algorithm (St Gallen 2013). Gene names associated with each subtype are presented in italics.

ESR1	PGR	ERBB2	MKI67/ Proliferation	Intrinsic Subtype
+	−	−	−	Luminal A-like
+	+	−	−	Luminal A-like
−	+	−	−	Luminal A-like
+	+	−	+	Luminal B-like (HER2 negative)
+	-	−	+	Luminal B-like (HER2 negative)
−	+	−	+	Luminal B-like (HER2 negative)
+	−	+	+	Luminal B-like (HER2 positive)
+	−	+	−	Luminal B-like (HER2 positive)
+	+	+	−	Luminal B-like (HER2 positive)
+	+	+	+	Luminal B-like (HER2 positive)
−	+	+	−	Luminal B-like (HER2 positive)
−	+	+	+	Luminal B-like (HER2 positive)
−	−	+	+	HER2 enriched (non-luminal)
−	−	+	−	HER2 enriched (non-luminal)
−	−	−	−	Triple Negative
−	−	−	+	Triple Negative

**Table 2 diagnostics-14-00241-t002:** Clinicopathological characteristics of the specimens used for analysis from cohort 1 and cohort 2. Results for ER, PR, HER2, and KI67 as listed in this table are reported based on immunohistochemistry.

Collection		Cohort 1		Cohort 2		
		N	%	N	%	
Material type						*p* = 0.70
	CNB	236	62.4	256	100.0	
	RES	142	37.6	0	0.0	
Histological type						*p* = 0.72
	Invasive ductal carcinoma	298	78.8	203	79.3	
	Invasive lobular carcinoma	56	14.8	38	14.8	
	Other	24	6.3	15	5.9	
Histological Grade						*p* = 0.58
	1	77	20.4	51	19.9	
	2	220	58.2	145	56.6	
	3	65	17.2	55	21.5	
	Unknown	16	4.3	5	2.0	
ER Status						*p* = 0.70
	Positive	311	82.3	190	74.2	
	Negative	67	17.7	66	25.8	
PR Status						*p* = 0.60
	Positive	281	74.3	163	63.7	
	Negative	97	25.7	93	36.3	
Ki67 (20% cut-off)						*p* = 0.62
	High	172	45.5	121	47.3	
	Low	205	54.2	134	52.3	
	Unknown	1	0.3	1	0.4	
HER2 Status						*p* = 0.76
	Positive	36	9.5	32	12.5	
	Negative	342	90.5	204	79.7	
	Unresolved	0	0.00	20	7.8	

**Table 3 diagnostics-14-00241-t003:** Contingency tables and agreement analysis for results generated by the APIS BC Subtyping Kit compared to results by IHC/ISH for each marker. Two *MKI67* and 20 *ERBB2* cases were excluded from the individual marker analysis due to lack of IHC and ISH data, respectively. Gene names and protein names are presented in italics and normal text, respectively.

Target		APIS Breast Cancer Subtyping Kit vs. IHC		
	Measure	N	%	95% CI
*ESR1*/ER	PPA (Sensitivity)	474/501	94.61	92.27–96.27
	NPA (Specificity)	117/133	87.97	81.35–92.46
	OPA	591/634	93.22	90.99–94.93
	PPV	474/490	96.73	94.76–97.98
	NPV	117/144	81.25	74.09–86.78
*PGR*/PR	PPA (Sensitivity)	405/444	91.22	88.22–93.51
	NPA (Specificity)	147/190	77.37	70.91–82.74
	OPA	552/634	87.07	84.23–89.46
	PPV	405/448	90.40	87.32–92.80
	NPV	147/186	79.03	72.62–84.27
*ERBB2*/HER2	PPA (Sensitivity)	62/68	91.18	82.06–95.89
	NPA (Specificity)	517/546	94.69	92.48–96.28
	OPA	579/614	94.30	92.18–95.87
	PPV	62/91	68.10	58.00–76.80
	NPV	517/523	98.90	97.50–99.50
*MKI67*/Ki67	PPA (Sensitivity)	234/293	79.86	74.90–84.06
	NPA (Specificity)	258/339	76.11	71.29–80.34
	OPA	492/632	77.85	74.45–80.91
	PPV	234/315	74.29	69.19–78.80
	NPV	258/317	81.39	76.74–85.29
*Proliferation*	PPA (Sensitivity)	233/293	79.52	74.53–83.75
	NPA (Specificity)	273/339	80.53	75.98–84.39
	OPA	506/632	80.06	76.77–82.99
	PPV	233/299	77.93	72.89–82.26
	NPV	273/333	81.98	77.50–85.74

**Table 4 diagnostics-14-00241-t004:** Contingency tables and agreement analysis for subtype calls generated by the APIS BC Subtyping Kit compared to calls made by pathologists based on IHC/ISH results.

Subtype		APIS Breast Cancer Subtyping Kit vs. Intrinsic Subtype		
	Measure	N	%	95% CI
Luminal A	PPA	232/314	73.89	68.76–78.43
	NPA	258/309	83.50	78.95–87.22
	OPA	490/623	78.65	75.26–81.69
Luminal B HER2−	PPA	96/152	63.16	55.25–70.41
	NPA	394/471	83.65	80.04–86.72
	OPA	490/623	78.65	75.26–81.69
Luminal B HER2+	PPA	27/32	84.38	68.25–93.14
	NPA	556/591	94.08	91.88–95.71
	OPA	583/623	93.58	91.38–95.25
HER2 Enriched	PPA	24/32	75.00	57.89–86.75
	NPA	585/591	98.98	97.80–99.53
	OPA	609/623	97.75	96.26–98.66
TNBC	PPA	67/93	72.04	62.19–80.15
	NPA	522/530	98.49	97.05–99.23
	OPA	589/623	94.54	92.47–96.07

**Table 5 diagnostics-14-00241-t005:** Contingency tables and agreement analysis for results from the APIS BC Subtyping Kit compared to results from IHC/ISH for each marker for the dataset as split by sample type (CNB and resected tissue). Gene names and protein names are presented in italics and normal text, respectively.

Target		CNB			Resected			
	Measure	N	%	95% CI	N	%	95% CI	
*ESR1*/ ER	Sensitivity	365/381	95.80	93.29–97.40	109/120	90.83	84.33–94.90	*p* = 0.55
	Specificity	98/111	88.29	80.99–93.03	19/22	86.36	66.67–95.25	*p* = 0.41
	PPV	365/378	96.56	94.21–97.98	109/112	97.32	92.42–99.08	*p* = 0.54
	NPV	98/114	85.96	78.41–91.17	19/30	63.33	45.51–78.13	*p* = 0.42
*PGR*/ PR	Sensitivity	314/340	92.35	89.03–94.73	91/104	87.50	79.78–92.55	*p* = 0.52
	Specificity	114//152	75.00	67.56–81.21	33/38	86.32	80.85–91.83	*p* = 0.29
	PPV	314/352	89.20	85.53–92.03	91/96	94.79	88.38–97.76	*p* = 0.47
	NPV	114/140	81.43	74.18–87.00	33/46	71.74	57.45–82.68	*p* = 0.41
*ERBB2*/ HER2	Sensitivity	58/64	90.63	81.02–95.63	8/9	88.89	56.50–98.01	*p* = 0.40
	Specificity	396/428	92.52	89.64–94.65	133/133	100.00	97.19–100.00	*p* = 0.53
	PPV	58/90	64.44	54.15–73.56	8/8	100.00	97.19–100.00	*p* = 0.09
	NPV	396/402	98.51	96.78–99.31	133/134	99.25	95.89–99.87	*p* = 0.58
*MKI67*/ Ki67	Sensitivity	204/236	86.44	81.49–90.23	30/57	52.63	39.92–65.01	*p* = 0.41
	Specificity	181/255	70.98	65.13–76.21	77/84	91.67	83.78–95.91	*p* = 0.31
	PPV	204/278	73.38	67.89–78.23	30/37	81.08	65.80–90.52	*p* = 0.21
	NPV	181/213	84.98	79.56–89.15	77/104	74.04	64.86–81.50	*p* = 0.56
*Proliferation*	Sensitivity	198/236	83.90	78.67–88.04	35/57	61.40	48.43–72.94	*p* = 0.38
	Specificity	198/255	77.65	72.14–82.33	75/84	89.29	80.88–94.26	*p* = 0.39
	PPV	198/255	77.65	72.14–82.33	35/44	79.55	65.55–88.85	*p* = 0.28
	NPV	198/236	83.90	78.67–88.04	75/97	77.32	68.04–84.52	*p* = 0.50

**Table 6 diagnostics-14-00241-t006:** ΔCt values and variability components attributed to each variable. Reported in terms of SD and %Total Variance. Analysis of data generated by Site 1—APIS Assay Technologies.

Target	Expression Level	Between- Operator		Between- Instrument		Between-Lot		Between-Run (Between Day)		Within Run		Total SD (100% Variance)
		% Total Variance	SD	% Total Variance	SD	% Total Variance	SD	% Total Variance	SD	% Total Variance	SD	
*ESR1*	Mid	48.459	0.344	0.400	0.031	0.000	0.000	22.085	0.232	29.055	0.266	0.494
	Low	9.673	0.136	10.349	0.141	14.406	0.166	19.765	0.195	45.807	0.297	0.438
	Negative	0.000	0.000	0.000	0.000	0.000	0.000	0.000	0.000	100.000	0.803	0.803
*PGR*	Mid	9.158	0.118	8.435	0.113	45.624	0.263	1.079	0.041	35.703	0.233	0.390
	Low	0.000	0.000	0.000	0.000	39.163	0.208	8.647	0.098	52.191	0.240	0.332
	Negative	5.557	0.154	0.000	0.000	0.000	0.000	0.000	0.000	94.443	0.635	0.654
*ERBB2*	Mid	40.459	0.191	0.000	0.000	5.492	0.070	18.258	0.128	35.791	0.180	0.301
	Low	0.000	0.000	10.498	0.063	0.714	0.016	51.230	0.139	37.558	0.119	0.194
	Negative	9.183	0.066	0.000	0.000	23.342	0.105	29.416	0.118	38.059	0.134	0.218
*MKI67*	Mid	0.000	0.000	10.845	0.096	32.167	0.165	0.000	0.000	56.989	0.220	0.291
	Low	0.000	0.000	11.552	0.091	23.485	0.129	4.668	0.058	60.294	0.207	0.267
	Negative	7.726	0.078	17.438	0.117	16.327	0.113	0.000	0.000	58.509	0.215	0.281
Proliferation Score	Mid	0.000	0.000	1.963	0.003	0.000	0.000	44.883	0.014	53.154	0.015	0.021
	Low	0.000	0.000	0.000	0.000	0.000	0.000	0.000	0.000	100.000	0.044	0.044
	Negative	38.062	0.035	10.269	0.018	3.452	0.011	1.395	0.007	46.822	0.039	0.057

**Table 7 diagnostics-14-00241-t007:** Total variance and standard deviation (SD) for ΔCt across all study factors (site, operator, instrument, lot, and run).

Target	Expression Level	Between-Site		Between- Operator		Between- Instrument		Between-Lot		Between-Run (Between Day)		Within Run		Total SD
		% Total Variance	SD	% Total Variance	SD	% Total Variance	SD	% Total Variance	SD	% Total Variance	SD	% Total Variance	SD	(100% Variance)
*ESR1*	Mid	10.948	0.223	25.288	0.339	16.07	0.27	2.801	0.113	25.768	0.342	19.125	0.295	0.674
	Low	0	0	21.888	0.338	0	0	46.999	0.495	10.083	0.229	21.03	0.331	0.722
	Negative	14.001	0.419	1.773	0.149	0	0	0	0	37.143	0.682	47.082	0.768	1.12
*PGR*	Mid	6.384	0.136	0.848	0.05	4.322	0.112	60.116	0.417	3.354	0.098	24.976	0.269	0.538
	Low	0	0	1.45	0.092	0	0	82.474	0.693	6.155	0.189	9.92	0.24	0.763
	Negative	6.989	0.173	1.003	0.065	0	0	0	0	0	0	92.008	0.626	0.653
*ERBB2*	Mid	0	0	4.664	0.127	2.628	0.095	71.026	0.497	13.712	0.218	7.971	0.166	0.589
	Low	9.25	0.156	0.127	0.018	1.028	0.052	73.721	0.441	2.907	0.088	12.968	0.185	0.514
	Negative	0	0	1.478	0.067	7.607	0.152	66.776	0.45	17.386	0.229	6.753	0.143	0.55
*MKI67*	Mid	0	0	1.007	0.057	1.061	0.059	69.09	0.476	1.576	0.072	27.265	0.299	0.573
	Low	1.348	0.06	3.662	0.099	1.807	0.07	60.749	0.405	3.586	0.098	28.849	0.279	0.52
	Negative	7.72	0.101	1.427	0.043	8.896	0.108	7.368	0.098	3.316	0.066	71.273	0.306	0.362
Proliferation Score	Mid	3.74	0.006	6.985	0.008	3.097	0.005	46.619	0.02	14.446	0.011	25.114	0.015	0.03
	Low	6.666	0.017	4.783	0.014	0	0	37.964	0.04	2.031	0.009	48.556	0.045	0.065
	Negative	25.196	0.04	38.062	0.035	10.269	0.018	3.452	0.011	1.395	0.007	46.822	0.039	0.081

**Table 8 diagnostics-14-00241-t008:** APIS BC Subtyping Kit binary precision—proportion of correct calls for each target and proliferation score across all sites.

Target	Expression Level	% Detection Rate
*ESR1*	Mid	100%
	Low	100%
	Negative	100%
*PGR*	Mid	100%
	Low	100%
	Negative	100%
*ERBB2*	Mid	100%
	Low	100%
	Negative	100%
*MKI67*	Mid	100%
	Low	100%
	Negative	87%
Proliferation Score	Mid	100%
	Low	100%
	Negative	99%

## Data Availability

The data presented in this study are available on request from the corresponding author. The data are not publicly available due to privacy.

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
