# Peer review of "Comprehensive and Accurate Molecular Profiling of Breast Cancer through mRNA Expression of ESR1, PGR, ERBB2, MKI67, and a Novel Proliferation Signature"

_diagnostics, 2024, doi:10.3390/diagnostics14030241_

Round 1
Reviewer 1 Report
Comments and Suggestions for Authors
Accurate determination of breast cancer biomarker status (ER, PR, HER2, Ki67) is known to be critical for patient management. The “gold standard” for assessing these biomarkers in FFPE tissues is IHC, which faces standardization challenges and exhibits significant variability. In the present study, the authors compared a new commercial RT-qPCR kit with IHC in determining BC biomarker status. It was shown that for all markers there was a high agreement with IHC (93.2% for ER, 87.1% for PR, 93.9% for HER2, 77.9% for Ki67 and 80.1% for the proliferative signature (assessed Ki67 IHC)).
1. Lines 196-201 - the information given in Table 3 is repeated, the text is difficult to read, it can be shortened.
2. Table 2 - when comparing 2 cohorts, it is necessary to provide a p-value for each indicator. Also in Table 5.
3. Were the IHC results assessed by a single pathologist? Or was there a double check?
Author Response
We would like to thank the reviewer for taking the time to review the manuscript.
|
Response to Reviewer 1 Comments
|
||
|
1. Summary |
|
|
|
We would like to thank the reviewer for taking the time to review this manuscript. Please find the detailed responses below and the corresponding revisions/corrections highlighted in yellow in the re-submitted files.
|
||
|
2. Questions for General Evaluation |
Reviewer’s Evaluation |
Response and Revisions |
|
Does the introduction provide sufficient background and include all relevant references? |
Yes |
Thank you for your evaluation |
|
Are all the cited references relevant to the research? |
Yes |
Thank you for your evaluation |
|
Is the research design appropriate? |
Yes |
Thank you for your evaluation |
|
Are the methods adequately described? |
Yes |
Thank you for your evaluation |
|
Are the results clearly presented? |
Can be improved |
Comments below have been addressed |
|
Are the conclusions supported by the results? |
Yes |
Thank you for your evaluation |
|
3. Point-by-point response to Comments and Suggestions for Authors |
||
|
Comments 1: Lines 196-201 - the information given in Table 3 is repeated, the text is difficult to read, it can be shortened.
|
||
|
Response 1: We agree with the reviewer’s comment. Therefore, we have amended the text to “The overall agreement between mRNA expression levels determined by APIS BC Subtyping Kit and IHC is shown in Table 3”. See page six, paragraph 3.1.2., lines 202-203 in the newly submitted manuscript. |
||
|
Comments 2: Table 2 - when comparing 2 cohorts, it is necessary to provide a p-value for each indicator. Also in Table 5. |
||
|
Response 2: We agree with the reviewer’s comment. We have added p-values to both Table 2 and Table 5 (see pages 5&6 and 9&10 of the newly submitted manuscript). |
||
|
Comments 3: Were the IHC results assessed by a single pathologist? Or was there a double check? |
||
|
Response 2: Thank you for pointing this out. We have amended the wording accordingly to clarify – “Tumor grading, tumor typing, and immunohistochemistry (ER, PR, HER2, Ki67) (IHC) were performed for all specimens by MVZ Prof. Dr. med. A. Niendorf Pathologie Hamburg-West GmbH. IHC was performed and analyzed by two board certified pathologists according to the institute’s standardized protocols and observing the ASCO/CAP guidelines [7] [8].”. See page 2, paragraph 2.1.1., lines 72-76. |
||
|
4. Response to Comments on the Quality of English Language |
||
|
Point 1: I am not qualified to assess the quality of English in this paper |
||
|
Response 1: N/A |
||
|
5. Additional clarifications |
||
|
N/A |
||
Reviewer 2 Report
Comments and Suggestions for Authors
The manuscript provides a comprehensive overview of the study, highlighting the need for accurate breast cancer biomarker determination and introducing the APIS BC Subtyping Kit as a potential solution. Here are some comments and suggestions for improvement:
1. Suggest prompting the authors to briefly mention any limitations associated with the study, such as sample biases or retrospective analysis. Additionally, encourage them to discuss potential future directions for research based on their findings.
2. Suggest advising the authors to include a few sentences in the Conclusion section that highlight the clinical relevance of their findings and potential implications for breast cancer diagnostics.
3. Suggest including a brief explanation or reference regarding how the clinically validated ΔCt cut-off values were determined for each marker.
4. Emphasize the clinical impact of the APIS BC Subtyping Kit. How might its use affect clinical decision-making, patient outcomes, or the overall management of breast cancer patients?
5.Encourage the authors to briefly discuss potential directions for future research based on the findings of this study.
6. Expand on the potential implications of the reproducibility and semi-quantitative capabilities of the APIS BC Subtyping Kit in terms of offering valuable opportunities for researchers. How could this contribute to uncovering meaningful associations between marker expression and treatment response?
Comments on the Quality of English Language
The manuscript is generally well-written. However, consider avoiding the repetition of phrases, such as "assessing the concordance with the results obtained through IHC" (mentioned in the Introduction and Conclusion).
Author Response
We would like to thank the reviewer for taking the time to review the manuscript.
|
Response to Reviewer 2 Comments
|
||
|
1. Summary |
|
|
|
We would like to thank the reviewer for taking the time to review this manuscript. Please find the detailed responses below and the corresponding revisions/corrections highlighted in yellow in the re-submitted files.
|
||
|
2. Questions for General Evaluation |
Reviewer’s Evaluation |
Response and Revisions |
|
Does the introduction provide sufficient background and include all relevant references? |
Yes |
Thank you for your evaluation |
|
Are all the cited references relevant to the research? |
Yes |
Thank you for your evaluation |
|
Is the research design appropriate? |
Can be improved |
Comments below have been addressed |
|
Are the methods adequately described? |
Can be improved |
Comments below have been addressed |
|
Are the results clearly presented? |
Can be improved |
Comments below have been addressed |
|
Are the conclusions supported by the results? |
Can be improved |
Comments below have been addressed |
|
3. Point-by-point response to Comments and Suggestions for Authors |
||
|
Comments 1: Suggest prompting the authors to briefly mention any limitations associated with the study, such as sample biases or retrospective analysis. Additionally, encourage them to discuss potential future directions for research based on their findings.
|
||
|
Response 1: Thank you for pointing this out. We agree with the reviewer’s comment and therefore modified the discussion section to emphasize points made. Limitations of the study can be found in lines 303-308 (“Our study's limitations include its retrospective design, relying on specimens pro-cured from a singular institution across two cohorts. Despite this, no substantial differences were noted between these cohorts. Cohort one was curated to maintain an even representation of all BC subtypes, whereas cohort two aimed to mirror the real-world distribution of BC subtypes, inadvertently yielding a greater number of HER2-negative specimens compared to HER2-positive specimens.”). Further research possibilities are now discussed in lines 317-320 (“Further exploration of the links between semi-quantitative mRNA expression and treatment response could yield valuable insights and improve the decision-making process, preventing potential over or under-treatment of patients.”), 326-330 (“Perhaps alternative RT-qPCR-based kits utilizing RNA extracted from the same tissue section could be used for comparison, helping to address potential discrepancies arising from tumor heterogeneity.”), 369-372 (“Understanding the correlation between APIS BC Subtyping Kit ERBB2 mRNA expression and response to anti-HER2 treatments could aid in stratifying patients into responder and non-responder groups, potentially reducing unnecessary severe side effects and offering a more personalized treatment approach” and 395-400 (“Further exploration of the prognostic potential of the APIS BC Subtyping Kit’s proliferation signature by evaluating its accuracy in predicting treatment outcomes and identifying subtypes alongside these commercially available tests could be of benefit. Given the costliness and centralized nature of these existing tests, a successful demonstration of a similar prognostic value by the APIS BC Subtyping Kit could position it as a decentralized, and accessible cost-effective alternative.”).
|
||
|
Comments 2: Suggest advising the authors to include a few sentences in the Conclusion section that highlight the clinical relevance of their findings and potential implications for breast cancer diagnostics.
|
||
|
Response 2: We agree with the reviewer’s comment. We amended the conclusions accordingly to emphasize these points. Additional information can be found in lines 427-431 (“The kit proved its reliability when used by clinical scientists in real clinical settings, exhibiting potential to improve the quality of primary breast cancer diagnostics. By providing results in a timely manner, the introduction of the APIS BC Subtyping Kit into routine practice in place of IHC could potentially save pathologist time and alleviate the pressure pathology laboratories currently face.”) and 440-442 (Furter research comparing the accuracy of APIS BC Subtyping Kit with different molecular tests, could provide useful insights into the clinical utility of molecular tests in clinical settings.”).
|
||
|
Comments 3: Suggest including a brief explanation or reference regarding how the clinically validated ΔCt cut-off values were determined for each marker.
|
||
|
Response 3: We agree with the reviewer’s comment. We have amended methods section (2.1.4) to include this information, see lines 109-113 – “ΔCt cut-off values were established by employing a binary classifier logistic regression model trained using the ΔCt expression data as input and the IHC status input as the bi-nary classifier. The value that achieved an equal balance between the sensitivity and specificity of the assay, and demonstrated the highest performance, was chosen as the ΔCt cut-off for each target.”
|
||
|
Comments 4: Emphasize the clinical impact of the APIS BC Subtyping Kit. How might its use affect clinical decision-making, patient outcomes, or the overall management of breast cancer patients?
|
||
|
Response 4: We agree with the reviewer’s comment. We have amended the discussion section accordingly to emphasize these points. See lines 297-302 (“The ability of the APIS BC Subtyping Kit to achieve highly repeatable and reproducible results is advantageous compared to current methods that rely on IHC and are prone to inconsistencies. The APIS BC Subtyping Kit minimizes the reliance on highly trained and experienced personnel for accurate results interpretation, thus improving the efficiency and reliability of breast cancer management.”), 350-354 (“Notably, the APIS BC Subtyping Kit provides a single-resolution ERBB2/HER2 detection method, removing the requirement for additional reflex testing – thus providing valuable information to healthcare providers in a rapid manner, reducing overall turnaround time and aiding fast patient management decisions.”), and 413-422 (“Once validated, the APIS BC Subtyping Kit would be able to provide information on individual biomarker status, BC subtype as well as recurrence risk in one test. This would distinguish it from the aforementioned tests, which don’t always report individual bi-omarker status, and therefore cannot be used for the initial biomarker assessment. Future studies could focus on comparing the subtype calls generated by the APIS BC Subtyping test and other molecular classifiers (e.g. Prosigna test). The APIS Breast Cancer Subtyping Kit can provide information about predictive and prognostic power of the examined parameters, especially on small biopsy material in, e.g., neoadjuvant settings planning cytotoxic chemotherapy or concerning proliferation rate in endocrine challenge (neoadjuvant endocrine therapy).”
|
||
|
Comments 5: Encourage the authors to briefly discuss potential directions for future research based on the findings of this study. |
||
|
Response 5: We agree with the reviewer’s comment. As discussed in the response to comment 1, the discussion section have now been amended to reflect further research. See lines 317-320 (“Further exploration of the links between semi-quantitative mRNA expression and treatment response could yield valuable insights and improve the decision-making process, preventing potential over or under-treatment of patients.”), 326-330 (“Perhaps alternative RT-qPCR-based kits already available on the market utilizing RNA extracted from the same tissue section could be used for comparison, helping to address potential discrepancies arising from tumor heterogeneity.”), 369-372 (“Understanding the correlation between APIS BC Subtyping Kit ERBB2 mRNA expression and response to anti-HER2 treatments could aid in stratifying patients into responder and non-responder groups, potentially reducing unnecessary severe side effects and offering a more personalized treatment approach” and 395-400 (“Further exploration of the prognostic potential of the APIS BC Subtyping Kit’s proliferation signature by evaluating its accuracy in predicting treatment outcomes and identifying subtypes alongside these commercially available tests could be of benefit. Given the costliness and centralized nature of these existing tests, a successful demonstration of a similar prognostic value by the APIS BC Subtyping Kit could position it as a decentralized, and accessible cost-effective alternative.”).
|
||
|
Comments 6: Expand on the potential implications of the reproducibility and semi-quantitative capabilities of the APIS BC Subtyping Kit in terms of offering valuable opportunities for researchers. How could this contribute to uncovering meaningful associations between marker expression and treatment response? |
||
|
Response 6: We agree with the reviewer’s comment. The discussion section has been amended to discuss the benefits of semi-quantitative approach. See lines 310-320 (“We found that the discrepancies were mainly seen in tumors with RNA levels close to the ∆Ct cut-offs and lower protein expression detected by IHC. This finding underscores the significance of employing semi-quantitative methods in result reporting, especially since previous studies indicated that low ER expressing tumors often present characteristics more similar to ER negative cancers [14]. As is still not clear what clinical implications these discrepancies may have, it is important to incorporate clinicopathological characteristics, such as grade, number of nodes and tumor size in patient management decisions [15]. Further exploration of the links between semi-quantitative mRNA expression and treatment response could yield valuable insights and improve the decision-making process, preventing potential over or under-treatment of patients.”), and 369-372 (“Understanding the correlation between APIS BC Subtyping Kit ERBB2 mRNA expression and response to anti-HER2 treatments could aid in stratifying patients into responder and non-responder groups, potentially reducing unnecessary severe side effects and offering a more personalized treatment approach.”)
|
||
|
4. Response to Comments on the Quality of English Language |
||
|
Point 1: The manuscript is generally well-written. However, consider avoiding the repetition of phrases, such as "assessing the concordance with the results obtained through IHC" (mentioned in the Introduction and Conclusion).
|
||
|
Response 1: We agree with the reviewer’s comment. Revisions have been made throughout the document to ensure no unnecessary repetitions.
|
||
|
5. Additional clarifications |
||
|
N/A |
||
Reviewer 3 Report
Comments and Suggestions for Authors
Wegscheider et al report on a mRNA-based ESR1, PgR, ERBB2 and Ki67 status in clinical breast cancer specimens, and derive intrinsic subtype assignments from these measurements as well as a proliferation signature from three additional genes. While a quantitative, precise method is clinically important given the ambiguity from IHC assessments in routine pathology, the value of the present study remains unclear as the methods are partly hidden or lack important information.
The following questions need to be addressed in a revised manuscript version:
-How were delta cT cutoffs defined for the four biomarkers?
-Table 1: on what basis/reference was subtype grouping done?
-Following the argument that IHC is error-prone, a better independent validation of subtype assignment based on mRNA would be a comparison to the PAM50 molecular classifier. Can you discuss please.
-Is the presented method of ct-value cutoff assignment entirely depending on the use of APIC kits and proprietary markers analysis? If so, I doubt that the method will replace current practice, as there are a number of alternatives available. This should be described/discussed
-The proliferation signature should be validated for its prognostic value and compared to established signatures such as PAM50 ROR score, Oncotype, Endopredict to support clinical utility. Any conclusions related to clinical value needs to be toned-down until these data are provided
Comments on the Quality of English LanguageLanguage is fine
Author Response
We would like to thank the reviewer for taking the time to review the manuscript.
|
Response to Reviewer 3 Comments
|
||
|
1. Summary |
|
|
|
We would like to thank the reviewer for taking the time to review this manuscript. Please find the detailed responses below and the corresponding revisions/corrections highlighted in yellow in the re-submitted files.
|
||
|
2. Questions for General Evaluation |
Reviewer’s Evaluation |
Response and Revisions |
|
Does the introduction provide sufficient background and include all relevant references? |
Yes |
Thank you for your evaluation |
|
Are all the cited references relevant to the research? |
Yes |
Thank you for your evaluation |
|
Is the research design appropriate? |
Can be improved |
Comments below have been addressed |
|
Are the methods adequately described? |
Must be improved |
Comments below have been addressed |
|
Are the results clearly presented? |
Yes |
Comments below have been addressed |
|
Are the conclusions supported by the results? |
Must be improved |
Comments below have been addressed |
|
3. Point-by-point response to Comments and Suggestions for Authors |
||
|
Comments 1: How were delta cT cutoffs defined for the four biomarkers? |
||
|
Response 1: We agree with the reviewer’s comment. We have amended methods section (2.1.4) to include this information, see lines 109-113 – “ΔCt cut-off values were established by employing a binary classifier logistic regression model trained using the ΔCt expression data as input and the IHC status input as the bi-nary classifier. The value that achieved an equal balance between the sensitivity and specificity of the assay, and demonstrated the highest performance, was chosen as the ΔCt cut-off for each target.”
|
||
|
Comments 2: Table 1: on what basis/reference was subtype grouping done? |
||
|
Response 2: We agree with the reviewer’s comment. We have now added reference to the source of this information. See lines 116-118 in section 2.1.4, page 2 (“The combination of ESR1, PGR, ERBB2 and MKI67 status was used to determine the molecular subtype, based on a definition by the St Gallen working group [12] (Table 1).”). Reference: [12] A. Goldhirsch, E. P. Winer, A. S. Coates , R. D. Gelber, M. Piccart-Gebhart, B. Thürlimann, H. J. Senn and Panel members, "Personalizing the treatment of women with early breast cancer: highlights of the St Gallen International Expert Consensus on the Primary Therapy of Early Breast Cancer 2013," Ann Oncol, vol. 24, no. 9, pp. 2206-23, 2013.
|
||
|
Comments 3: Following the argument that IHC is error-prone, a better independent validation of subtype assignment based on mRNA would be a comparison to the PAM50 molecular classifier. Can you discuss please. |
||
|
Response 3: We agree with the reviewer’s comment. We have accordingly modified the discussion section (Section 4). See lines 387-400 (“Various multigene tests addressing this issue exist. Oncotype Dx® (Exact Sciences), Prosigna® (NanoString Technologies), EndoPredict® (Myriad Genetics) and MammaPrint® (Agendia) provide risk scores with prognostic information for distant recurrence in patients with Luminal A and Luminal B tumors. The risk determined by these tests (high/low) is often used to guide chemotherapy treatment decision [27]. However, these tests are often centralized and very expensive, and therefore only used as an additional test for those cases that require further assessment. Further exploration of the prognostic potential of the APIS BC Subtyping Kit’s proliferation signature by evaluating its accuracy in predicting treatment outcomes and identifying subtypes alongside these commercially available tests could be of benefit. Given the costliness and centralized nature of these existing tests, a successful demonstration of a similar prognostic value by the APIS BC Subtyping Kit could position it as a decentralized, and accessible cost-effective alternative.”) and 413-422 (“Once validated, the APIS BC Subtyping Kit would be able to provide information on individual biomarker status, BC subtype as well as recurrence risk in one test. This would distinguish it from the aforementioned tests, which don’t always report individual biomarker status, and therefore cannot be used for the initial biomarker assessment. Future studies could focus on comparing the subtype calls generated by the APIS BC Subtyping test and other molecular classifiers (e.g. Prosigna test). The APIS Breast Cancer Subtyping Kit can provide information about predictive and prognostic power of the examined parameters, especially on small biopsy material in, e.g., neoadjuvant settings planning cytotoxic chemotherapy or concerning proliferation rate in endocrine challenge (neoadjuvant endocrine therapy).”)
|
||
|
Comments 4: Is the presented method of ct-value cutoff assignment entirely depending on the use of APIC kits and proprietary markers analysis? If so, I doubt that the method will replace current practice, as there are a number of alternatives available. This should be described/discussed |
||
|
Response 4: We agree with the reviewer’s comment. Currently on the market there are other kits providing similar information to APIS BC Subtyping Kit. These kits also utilize their proprietary method for marker analysis with independently validated cut-off values. We amended the discussion section to include the mention of those tests (see lines 326-330 “Perhaps alternative RT-qPCR-based kits already available on the market utilizing RNA extracted from the same tissue section could be used for comparison, helping to address potential discrepancies arising from tumor heterogeneity.”) and emphasized the benefits of APIS BC Subtyping Kit over alternative methods (see lines 297-302 “The ability of the APIS BC Subtyping Kit to achieve highly repeatable and reproducible results is advantageous compared to current methods that rely on IHC and are prone to inconsistencies. The APIS BC Subtyping Kit minimizes the reliance on highly trained and experienced personnel for accurate results interpretation, thus improving the efficiency and reliability of breast cancer management.”, and 350-354 “Notably, the APIS BC Subtyping Kit provides a single-resolution ERBB2/HER2 detection method, removing the requirement for additional reflex testing – thus providing valuable information to healthcare providers in a rapid manner, reducing overall turnaround time and aiding fast patient management decisions.”)
|
||
|
Comments 5: The proliferation signature should be validated for its prognostic value and compared to established signatures such as PAM50 ROR score, Oncotype, Endopredict to support clinical utility. Any conclusions related to clinical value needs to be toned-down until these data are provided |
||
|
Response 5: We agree with the reviewer’s comment. We added consideration about comparing the performance of APIS BC Subtyping Kit to PAM50, Oncotype and Endopredict to the discussions section as described in the response to comment 3. We ensured to make it clear that prognostic value of the kit has not yet been validated (lines 413-414 of the discussion section “Once validated, the APIS BC Subtyping Kit would be able to provide information on individual biomarker status, BC subtype as well as recurrence risk in one test.”) and ensured the conclusion sections mentions future efforts to validate the prognostic potential, implying this has not been performed yet (lines 435-437 “Further clinical investigations are required to validate the APIS BC Subtyping Kit’s prognostic potential and establish appropriate thresholds for semi-quantitative evaluation of ERBB2/HER2.”). Currently, a multicenter study is being planned to validate these findings.
|
||
|
4. Response to Comments on the Quality of English Language |
||
|
Point 1: Language is fine |
||
|
Response 1: We would like to thank the reviewer for the comment. |
||
|
5. Additional clarifications |
||
|
N/A |
||
Round 2
Reviewer 3 Report
Comments and Suggestions for Authors
The revision is acceptable yet require some additional language improvements. Specifically, given the clinical value of individual mRNA counts and the proliferation score for prognosis are not tested here, I recommend to point to this limitation more directly and avoid indirect (vague) phrases.
Specific
Line 317 ff: Further exploration of the links between semi-quantitative mRNA expression and treatment response (delete: could yield valuable insights and improve) ARE REQUIRED TO DEMONSTRATE ITS CLINICAL VALUE FOR THERAPEUTIC decision-making.
Line 326 ff: (Delete: Perhaps) alternative..
Line 394 ff: (Delete: Further) THE Exploration of the prognostic potential of the APIS BC Subtyping Kit’s proliferation signature by TESTING its UTILITY in predicting treatment outcomes and identifying subtypes IN COMPARISON TO the available tests WILL BE IMPORTANT (DELETE: Given the costliness and centralized nature of these existing tests, a successful demonstration of a similar prognostic value by the APIS BC Subtyping Kit could position it as) TO PROVIDE a decentralized, and accessible cost-effective alternative.
Line 412 ff: Once validated, the APIS BC Subtyping Kit (Delete: would be able) COULD provide..
Line 415 ff: Future studies (Delete: could focus) ARE REQUIRED TO comparE the subtype calls generated by the APIS BC Subtyping test and other molecular classifiers (e.g. Prosigna test).
Line 427 ff: By providing results in a timely manner, the (Delete: introduction of the) APIS BC Subtyping Kit HOLDS POTENTIAL TO IMPROVE routine practice (DELETE: in place of IHC could potentially) BY saveING pathologist time (DELETE: alleviate the pressure pathology
laboratories currently face.)
Line 439 ff: Delete whole sentence (Furter research comparing the accuracy of APIS BC Subtyping Kit with different molecular tests, could provide useful insights into the clinical utility of molecular tests in clinical settings.) as is already said.
Comments on the Quality of English LanguageSee comments above
Author Response
We'd like to thank the reviewer for all comments.
For research article
|
Response to Reviewer 3 Comments
|
||
|
1. Summary |
|
|
|
We would like to thank the reviewer for taking the time to review this manuscript. Please find the detailed responses below and the corresponding revisions/corrections highlighted in yellow in the re-submitted files.
|
||
|
2. Questions for General Evaluation |
Reviewer’s Evaluation |
Response and Revisions |
|
Does the introduction provide sufficient background and include all relevant references? |
Yes |
Thank you for your evaluation |
|
Are all the cited references relevant to the research? |
Yes |
Thank you for your evaluation |
|
Is the research design appropriate? |
Yes |
Thank you for your evaluation |
|
Are the methods adequately described? |
Yes |
Thank you for your evaluation |
|
Are the results clearly presented? |
Yes |
Thank you for your evaluation |
|
Are the conclusions supported by the results? |
Can be improved |
Comments below have been addressed |
|
3. Point-by-point response to Comments and Suggestions for Authors |
||
|
Comments 1: Line 317 ff: Further exploration of the links between semi-quantitative mRNA expression and treatment response (delete: could yield valuable insights and improve) ARE REQUIRED TO DEMONSTRATE ITS CLINICAL VALUE FOR THERAPEUTIC decision-making. |
||
|
Response 1: We agree with the reviewer’s comment. We have amended the sentence which now reads: “Further exploration of the links between semi-quantitative mRNA expression and treatment response are required to demonstrate its clinical value for therapeutic decision-making.” (Lines 317-319). |
||
|
Comments 2: Line 326 ff: (Delete: Perhaps) alternative… |
||
|
Response 2: We agree with the reviewer’s comment. We have amended the sentence which now reads: “Alternative RT-qPCR-based kits already available on the market utilizing RNA extracted from the same tissue section could be used for comparison, helping to address potential discrepancies arising from tumor heterogeneity.” (Lines 325-328) |
||
|
Comments 3: Line 394 ff: (Delete: Further) THE Exploration of the prognostic potential of the APIS BC Subtyping Kit’s proliferation signature by TESTING its UTILITY in predicting treatment outcomes and identifying subtypes IN COMPARISON TO the available tests WILL BE IMPORTANT (DELETE: Given the costliness and centralized nature of these existing tests, a successful demonstration of a similar prognostic value by the APIS BC Subtyping Kit could position it as) TO PROVIDE a decentralized, and accessible cost-effective alternative. |
||
|
Response 3: We agree with the reviewer’s comment. We have amended the sentence which now reads: “The exploration of the prognostic potential of the APIS BC Subtyping Kit’s proliferation signature by testing its utility in predicting treatment outcomes and identifying subtypes in comparison to the commercially available tests will be important to provide a centralized, and accessible cost-effective alternative.” (Lines 393-396) |
||
|
Comments 4: Line 412 ff: Once validated, the APIS BC Subtyping Kit (Delete: would be able) COULD provide… |
||
|
Response 4: We agree with the reviewer’s comment. We have amended the sentence which now reads: “Once validated, the APIS BC Subtyping Kit could provide information on individual biomarker status, BC subtype as well as recurrence risk in one test.” (Lines 409-410) |
||
|
Comments 5: Line 415 ff: Future studies (Delete: could focus) ARE REQUIRED TO compare the subtype calls generated by the APIS BC Subtyping test and other molecular classifiers (e.g. Prosigna test). |
||
|
Response 5: We agree with the reviewer’s comment. We have amended the sentence which now reads: “Future studies are required to compare the subtype calls generated by the APIS BC Subtyping test and other molecular classifiers (e.g. Prosigna test).“ (Lines 412-414) |
||
|
Comments 6: Line 427 ff: By providing results in a timely manner, the (Delete: introduction of the) APIS BC Subtyping Kit HOLDS POTENTIAL TO IMPROVE routine practice (DELETE: in place of IHC could potentially) BY saveING pathologist time (DELETE: alleviate the pressure pathology laboratories currently face.) |
||
|
Response 6: We agree with the reviewer’s comment. We have amended the sentence which now reads: “By providing results in a timely manner, the APIS BC Subtyping Kit holds potential to improve routine practice, by saving pathologists time.” (Lines 424-426) |
||
|
Comments 7: Line 439 ff: Delete whole sentence (Furter research comparing the accuracy of APIS BC Subtyping Kit with different molecular tests, could provide useful insights into the clinical utility of molecular tests in clinical settings.) as is already said. |
||
|
Response 7: We agree with the reviewer’s comment. The sentence has been deleted. |
||
|
4. Response to Comments on the Quality of English Language |
||
|
Point 1: See comments above |
||
|
Response 1: The comments have been addressed |
||
|
5. Additional clarifications |
||
|
N/A |
||